# Mismatch Repair Status in Patient-Derived Colorectal Cancer Organoids Does Not Affect Intrinsic Tumor Cell Sensitivity to Systemic Therapy

**DOI:** 10.3390/cancers13215434

**Published:** 2021-10-29

**Authors:** Emre Küçükköse, G. Emerens Wensink, Celine M. Roelse, Susanne J. van Schelven, Daniëlle A. E. Raats, Sylvia F. Boj, Miriam Koopman, Jamila Laoukili, Jeanine M. L. Roodhart, Onno Kranenburg

**Affiliations:** 1Laboratory Translational Oncology, Division of Imaging and Cancer, University Medical Center Utrecht, 3584 CX Utrecht, The Netherlands; e.kucukkose@umcutrecht.nl (E.K.); c.roelse@umcutrecht.nl (C.M.R.); s.j.vanschelven@umcutrecht.nl (S.J.v.S.); d.a.e.raats@umcutrecht.nl (D.A.E.R.); j.laoukili@umcutrecht.nl (J.L.); 2Division of Imaging and Cancer, Department of Medical Oncology, University Medical Center Utrecht, 3584 CX Utrecht, The Netherlands; g.e.wensink@umcutrecht.nl (G.E.W.); m.koopman-6@umcutrecht.nl (M.K.); 3Foundation Hubrecht Organoid Technology, 3584 CM Utrecht, The Netherlands; s.boj@hub4organoids.nl; 4Utrecht Platform for Organoid Technology, Utrecht University, 3584 CX Utrecht, The Netherlands

**Keywords:** organoids, colorectal cancer, deficient mismatch repair, mutation, drug screens

## Abstract

**Simple Summary:**

The large majority of patients with metastatic colorectal cancer (mCRC) receive chemotherapy with or without targeted therapy. Tumors with a deficient DNA mismatch repair (dMMR) system respond poorly to systemic therapy and are associated with poor prognosis. However, it is unclear whether dMMR causes therapy resistance in a tumor cell-intrinsic manner, or whether other mechanisms underlie this association. We address this issue by exposing a panel of MMR-deficient and -proficient Patient-Derived Organoids (PDOs) to a series of clinically relevant drugs. We show that MMR status did not correlate with the response of PDOs to any of the drugs tested. By contrast, the presence of activating mutations in the *KRAS* and *BRAF* oncogenes was significantly associated with resistance to chemotherapy and sensitivity to drugs targeting oncogene-activated pathways. We conclude that tumor cell-intrinsic signals link oncogene status, but not MMR status, to variation in therapy response in CRC.

**Abstract:**

DNA mismatch repair deficiency (dMMR) in metastatic colorectal cancer (mCRC) is associated with poor survival and a poor response to systemic treatment. However, it is unclear whether dMMR results in a tumor cell-intrinsic state of treatment resistance, or whether alternative mechanisms play a role. To address this, we generated a cohort of MMR-proficient and -deficient Patient-Derived Organoids (PDOs) and tested their response to commonly used drugs in the treatment of mCRC, including 5-fluorouracil (5-FU), oxaliplatin, SN-38, binimetinib, encorafenib, and cetuximab. MMR status did not correlate with the response of PDOs to any of the drugs tested. In contrast, the presence of activating mutations in the *KRAS* and *BRAF* oncogenes was significantly associated with resistance to chemotherapy and sensitivity to drugs targeting oncogene-activated pathways. We conclude that mutant KRAS and BRAF impact the intrinsic sensitivity of tumor cells to chemotherapy and targeted therapy. By contrast, tumor cell-extrinsic mechanisms—for instance signals derived from the microenvironment—must underlie the association of MMR status with therapy response. Future drug screens on rationally chosen cohorts of PDOs have great potential in developing tailored therapies for specific CRC subtypes including, but not restricted to, those defined by BRAF/KRAS and MMR status.

## 1. Introduction

A deficient DNA mismatch repair (dMMR) system underlies the formation of a specific subtype of colorectal cancer (CRC), accounting for approximately 15% of all cases of primary CRC. In general, dMMR is associated with a lower risk of distant metastasis formation and a relatively good prognosis [1]. However, a minority of patients with dMMR CRC do develop metastatic disease. This group, making up 3–5% of all cases of metastatic CRC (mCRC), is associated with an unfavorable prognosis [2,3,4], which has been linked to a reduced sensitivity to systemic treatment [5]. Indeed, patients with metastatic dMMR tumors receiving chemotherapy and targeted treatment have a lower response rate (5% versus 44%) and a shorter progression-free survival, when compared to patients with proficient-MMR tumors (pMMR) [6,7]. Moreover, patients with non-metastatic dMMR CRC benefit less from adjuvant treatment with 5-FU monotherapy than patients with pMMR CRC [8,9,10].

Immune checkpoint inhibitors (ICI) have revolutionized the treatment of patients with dMMR mCRC, producing durable responses that result in a significantly improved survival [11]. Despite the success of ICI therapy, approximately one third of the patients receiving such treatment display resistance [11]. For those patients, chemotherapy and targeted treatment remain a relevant (often the only) treatment option. However, all mCRC patients receive the same (non-immunotherapy) systemic treatment, regardless of MMR status. Since dMMR status is associated with a worse prognosis and response rate, it is imperative to understand the basis of this differential outcome. Eventually, such knowledge may allow the design of more effective treatment strategies that are based on patient stratification according to MMR status [5].

Organoid technology can be used to generate Patient-Derived tumor Organoid (PDO) cultures, in which the genetic and functional characteristics of the original tumor are maintained [12,13,14]. Importantly, clinical responses of colorectal tumors to systemic therapy are faithfully reproduced in PDO-based drug screens in vitro [13,14,15]. In the present study, we generated a series of genetically characterized PDOs from dMMR and pMMR CRC to assess whether tumor cell-intrinsic parameters could explain the clinically observed differences in response to systemic therapy between these CRC subgroups. The generated drug response data were also compared to the presence of driver mutations in *KRAS* and *BRAF*.

## 2. Materials and Methods

### 2.1. Human Specimens

Tissue samples from CRC patients were collected during surgery, or by fine-needle biopsies, within the Biobanking protocol HUB-Cancer TCBIO #12-093, which was approved by the medical ethical committee of the University Medical Center Utrecht (UMCU). Written informed consent from the donors was obtained prior to acquisition of the specimen for research use in the present study.

### 2.2. In Vitro Organoid Culture

Culturing organoids was performed by embedding in ice-cold Matrigel^®^ (Corning, Corning, NY, USA), mixed with a CRC culture medium (Appendix A) in a 3:1 ratio. For passaging, the tumor organoids were dissociated with TrypLE Express (Gibco, Breda, The Netherlands, #12604021) for 5–10 min at 37 °C and re-plated in a pre-warmed 6-well plate. Rho-associated kinase (ROCK) inhibitor Y-27632 (Tocris, Abingdon, UK, #1254, 10 μM) was added to culture medium upon plating for two days.

### 2.3. Immunohistochemistry (IHC)

PDOs were cultured for 10 days and harvested using 2 mg/mL Dispase type II (Sigma-Aldrich, Zwijndrecht, The Netherlands, #D4693) by 15 min incubation at 37 °C. After washout of Dispase type II with PBS (Sigma-Aldrich, Zwijndrecht, The Netherlands), the organoids were fixed in 4% formaldehyde solution at room temperature for 20 min. The formaldehyde solution was aspirated and 200 µL 2% Agar (Merck, Kenilworth, NJ, USA) solution was added and hardened on a pre-cooled dish. Subsequently, the agar-droplets containing PDOs were embedded in paraffin blocks. For assessment of mismatch-repair status, immunohistochemistry staining was performed on a BenchMark Ultra Autostainer (Ventana Medical Systems, Oro Valley, AZ, USA). Briefly, paraffin sections were cut at 4 µm and deparaffinized in the instrument with EZ prep solution (Ventana Medical Systems) at 75 °C for 8 min. Heat-induced antigen retrieval was carried out using Cell Conditioning 1 (CC1, Ventana Medical Systems) for 32 min at 100 °C. The following antibodies were used: anti-hMLH1 (BD Pharmingen Amsterdam, The Netherlands, G168-15, 1:20), anti-hPMS2 (Roche, Woerden, The Netherlands, EPR3947, ready-to-use), anti-hMSH2 (Roche, G219-1129, ready-to-use), and anti-hMSH6 (Abcam, Cambridge, UK, EPR3945, 1:200). Slides were counterstained with Hematoxylin and Bluing Reagent (Ventana Medical Systems).

### 2.4. DNA Isolation and Whole-Genome-Sequencing (WGS)

PDOs were harvested as described previously in 2.3. Total DNA was isolated using the DNeasy Mini Kit (Qiagen, Hilden, Germany) according to the manufacturer’s instructions. Extracted genomic material concentration and quality was measured using NanoDrop 2000 (Thermo Scientific, Waltham, MA, USA) and Bioanalyzer 2100 (Agilent, Santa Clara, CA, USA). Whole genome sequencing of the PDOs was performed on an Illumina NovaSeq 6000 (2 × 150 bp). Mapping and assessment of germline single-nucleotide-variation (SNV) and insertion and/or deletion variants was performed using the “Illumina Analysis Pipeline” of Utrecht Bioinformatics Expertise Core within the Center for Molecular Medicine at the UMC.

### 2.5. Drug Screens

All drug screens were performed twice on different days. PDOs were mechanically and enzymatically dissociated into single cells by incubating in TrypLE for 5–10 min and replated to allow for the formation of organoids over the course of 72 h. After 72 h, PDOs were collected, incubated with Dispase II (Sigma-Aldrich, #D4693, 2 mg/mL) for 15 min at 37 °C to remove Matrigel, and counted using a hemocytometer and trypan blue. PDOs (50 organoids/μL) were resuspended in 13 mL CRC culture medium without NAC and with low EGF concentration (1 ng/mL) supplemented with 5% Matrigel. PDO suspension (40 μL/well) was dispensed in clear-bottomed, black-walled 384-well plates (Corning, #3904) using an automated Multidrop™ Combi Reagent Dispenser. We generated a twelve-step, threefold drug matrices of 5-FU, oxaliplatin, SN-38, cetuximab, encorafenib, and binimetinib, as described in Appendix A using a Tecan D300e digital dispenser. Readouts were obtained at day six in empty wells, positive control (20 μg/mL puromycin), negative control (1% Dimethyl sulfoxide or 0.3% Tween-20, depending on solvent of agent used), and the drug-treated wells [16]. Quantification was done on a SpectraMax plate reader (Molecular Devices) by measuring cell viability using CellTiter-Glo 3D (Promega, #G9681, 40 µL/well) according to the manufacturer’s instructions. Screens with strictly standardized mean difference (SSMD) values <3.0 were classified as poor quality and excluded from the analysis [17].

### 2.6. Dose–Response Curve (DRC) Fitting and Statistical Analysis

Drug responses were calculated on the basis of normalized viability data as previously described [16], and were fitted by log10-transformed drug concentrations using a 4-parameter logistic regression model using the nplr package (v.0.1-7) in R. Nonmonotonic DRCs of 5-FU and encorafenib were fitted using Dr-fit software [18]. Encorafenib’s maximum concentration was adjusted after initial analysis at 1 µM due to biphasic stimulatory effect on curve fitting at higher concentrations, presumably due to complex formation.

One DRC parameter was analyzed: area under the curve (AUC), inferred by integrating fitted curves using Simpson’s rule. To analyze the reproducibility between drug screens, we calculated Pearson’s R and corresponding *p*-value using the response values in the drug matrix. Pre-specified CRC subgroups (MMR status and *KRAS*/*BRAF* mutational-status) were compared using a Mann–Whitney U-test or Kruskal–Wallis test. *p*-values of <0.05 were considered significant.

## 3. Results

### 3.1. A Collection of dMMR and pMMR CRC Patient-Derived Organoids

To study the tumor cell-intrinsic differences in response to commonly used therapies between dMMR and pMMR CRC, we generated a PDO cohort of six dMMR and six pMMR CRC-derived PDOs with annotated primary tumor location (sidedness) and *BRAF*- and *KRAS*-mutational status (Table 1). The MMR status of all 12 PDOs was assessed by immunohistochemistry analysis of the expression of the MMR proteins MLH1, PMS2, MSH2, and MSH6 (Figure 1). All six pMMR PDOs had normal expression of all MMR proteins. By contrast, five (out of six) dMMR PDOs displayed loss of MLH1 and PMS2 expression, and one PDO (dMMR3) displayed loss of MSH2/MSH6 expression. Whole-genome (DNA) sequencing revealed that dMMR3 and dMMR5 had a loss of function mutation in *MSH2* (p.Glu483*), and inactivating mutations in *MLH1* (p.I219V) and *MSH3* (p.W1111R), indicating that these PDOs were derived from patients with Lynch syndrome (Figure 2).

### 3.2. Comparable Sensitivity of dMMR and pMMR PDOs to Chemotherapy and Targeted Therapy

All PDOs were screened for their sensitivity to commonly used drugs in the treatment of metastatic CRC, including the chemotherapeutic drugs 5-FU, oxaliplatin and irinotecan (SN-38), the EGFR-targeting antibody cetuximab, the BRAF inhibitor encorafenib, and the MEK inhibitor binimetinib (Figure 3A). All 12 PDOs were exposed to concentration series of all drugs for a period of three days. Cell Titer Glo was then used to measure ATP levels as a proxy for the amount of viable (metabolically active) cells. Drug effects were calculated by normalizing the recorded values against those from drug solvent-exposed control wells (Appendix A). The quality of all drug screens was analyzed using the strictly standardized mean difference (SSMD) and the observed correlation between replicates. Screens were excluded from subsequent analyses if they had an SSMD < 3.0. The SSMD of all assays for all PDOs are shown in Appendix A. Based on these quality assessments, six drug screens were excluded and 24 screens were included in the subsequent analyses. All PDOs were screened against all drugs twice on different days. The normalized cell viabilities of the biological triplicates of the analyzed assays were significantly correlated (*Pearson’ R* = 0.73 − 0.76; *p* < 0.05, Appendix A). In addition, the variation was minimal (1.09%) between the duplicate assays of each PDO (Appendix A).

To investigate whether MMR status correlated with the sensitivity of CRC PDOs to any of the tested drugs, we generated dose response curves (DRC) and calculated the area under the curve (AUC) values for each of the fitted curves. We found no significant differences in sensitivity to any of the drugs between the groups of dMMR and pMMR PDOs (Figure 3B,C and Appendix A). Thus, as a group, dMMR PDOs are not intrinsically more or less sensitive to chemotherapy and targeted treatment when compared to pMMR PDOs in vitro.

### 3.3. Association of BRAF and KRAS Status with Response to Chemotherapy and Targeted Therapy

While the study was designed to compare PDOs from dMMR and pMMR CRC, additional information on the PDO cohort allowed us to also compare drug responses between subgroups differing in driver gene mutation status (*BRAF*, *KRAS*). These analyses revealed that *BRAF*- and *KRAS*-mutant PDOs were significantly (*p* = 0.017 and *p* = 0.008) more resistant to 5-FU and oxaliplatin when compared to *BRAF*/*KRAS* wildtype PDOs (Figure 4A and Appendix A). By contrast, *BRAF*/*KRAS* status had no significant effect on the sensitivity of PDOs to irinotecan/SN-38 (Figure 4A).

*BRAF*-mutant PDOs showed a significantly higher sensitivity to the BRAF inhibitor encorafenib than *KRAS*-mutant or *BRAF*/*KRAS* wildtype PDOs (Figure 4B). Both *BRAF*- and *KRAS*-mutated PDOs displayed a significantly increased sensitivity to the MEK inhibitor binimetinib when compared to wildtype PDOs. There was no significant difference in cetuximab sensitivity among PDOs based on *BRAF/KRAS* mutational status (Figure 4B). This is perhaps surprising, given the clinical correlation between the presence of mutations in *KRAS* and *BRAF* with resistance to cetuximab [19,20,21]. However, one of the cetuximab-resistant PDOs (dMMR2) with wildtype KRAS and BRAF did contain an activating mutation in ERBB3, which is also known to confer resistance to EGFR inhibition [22] (Figure 2).

## 4. Discussion

The data presented in this study show that MMR status does not affect the intrinsic sensitivity of PDOs to chemotherapy or targeted therapy. The poor prognosis of dMMR mCRC patients can therefore not simply be explained by a reduced sensitivity of dMMR tumor cells to systemic therapy. An alternative explanation could be that the distinct microenvironments of pMMR and dMMR CRC could influence treatment response [23]. Addressing this experimentally would require the generation of PDO co-culture models in which these distinct microenvironments are modeled, for instance by co-culturing PDOs with specific immune cell types, or by transplanting PDOs into mice with a human immune system [24,25]. Insight into the mechanisms underlying the poor clinical response of dMMR mCRC to systemic therapy is essential in order to design more effective treatment strategies. Until such mechanistic insight is provided, the only available treatment options for immunotherapy-refractory dMMR mCRC remain standard chemotherapy schedules with or without targeted therapy. Some studies indicate that MMR status may be associated with an increased or reduced benefit from specific treatment schedules, but solid evidence for such relationships is currently lacking [26,27,28,29].

This study was designed to study the effect of MMR status on the tumor cell-intrinsic responsiveness to a series of clinically relevant drugs. A potentially relevant variable that we were unable to address in this study was the origin of the dMMR phenotype: sporadic versus inherited (Lynch syndrome). Likewise, tumor sidedness is a potentially important variable in determining treatment responses. Studies with specifically selected (or generated) PDO cohorts (i.e., sporadic dMMR versus Lynch syndrome; left-sided versus right sided, etc.) are required to address these questions in the future.

Many studies have found correlations between the presence of specific genetic alterations (including but not limited to mutations in *KRAS* and *BRAF*) with resistance to chemotherapy. However, none of these are currently used to select patients for, or exclude them from, treatment with chemotherapy [30]. By contrast, specific genetic alterations are used to select patients for targeted therapy. Examples are the selection of patients with mutant *BRAF* tumors for treatment with BRAF inhibitor combination therapies, the selection of patients with dMMR tumors for immunotherapy, and the exclusion of patients with *KRAS* mutant tumors from treatment with EGFR-targeting therapy [30]. Although this study was not designed to study a potential effect of mutant *BRAF* and *KRAS* on treatment response, we found that within the PDO cohort used, the presence of these oncogenes was significantly associated with reduced sensitivity to 5-FU and oxaliplatin. An intrinsic (relative) resistance of tumor cells with mutant *BRAF* or *KRAS* to chemotherapy may contribute to the poor prognosis that is associated with the presence of these oncogenes [31,32]. The data also show that mutant *BRAF* PDOs are more sensitive to inhibitors of BRAF and that *KRAS* and *BRAF* mutant PDOs are more sensitive to inhibition of the BRAF target MEK. The results therefore support the concept of targeted inhibition of oncogene-activated signaling pathways as a basis for future treatment strategies, involving the concomitant suppression of feedback pathways.

## 5. Conclusions

By using PDOs as a model system for evaluating clinically relevant drug responses, we conclude that KRAS and BRAF, but not MMR status, are dominant factors in determining the intrinsic sensitivity of tumor cells to chemotherapy and targeted therapy. Future drug screens on rationally chosen cohorts of PDOs have great potential in developing tailored therapies for specific CRC subtypes including, but not restricted to, those defined by BRAF/KRAS and MMR status.

## Figures and Tables

**Figure 1 cancers-13-05434-f001:**
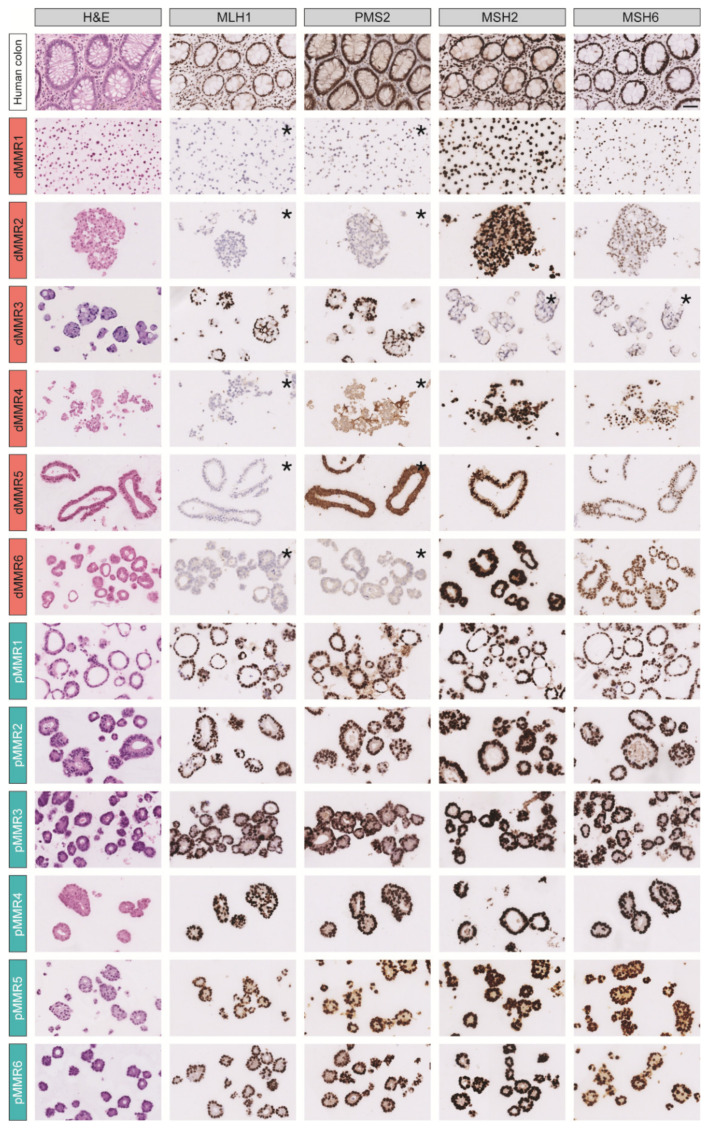
Expression of mismatch repairs proteins (MLH1, PMS2, MSH2 and MSH6) in PDOs analyzed by immunohistochemistry. * demonstrates loss of expression. Scale bar shown in human colon MSH6 staining is 50 µm and applies for all images.

**Figure 2 cancers-13-05434-f002:**
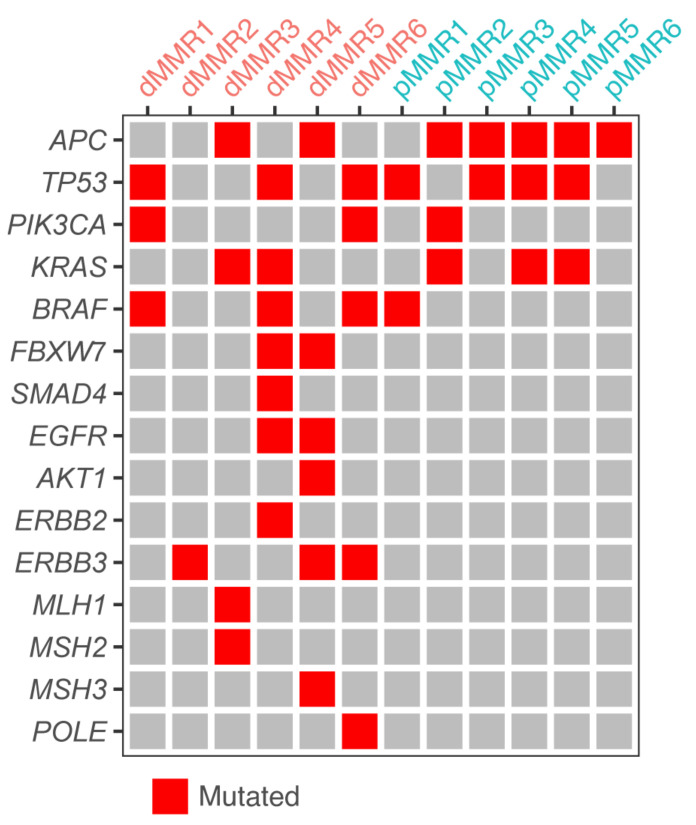
Mutational status of genes frequently associated with CRC. Overview of all PDOs with mutations in genes commonly mutated in CRC. “Mutated” was defined as a given variant being predicted pathogenic by COSMIC.

**Figure 3 cancers-13-05434-f003:**
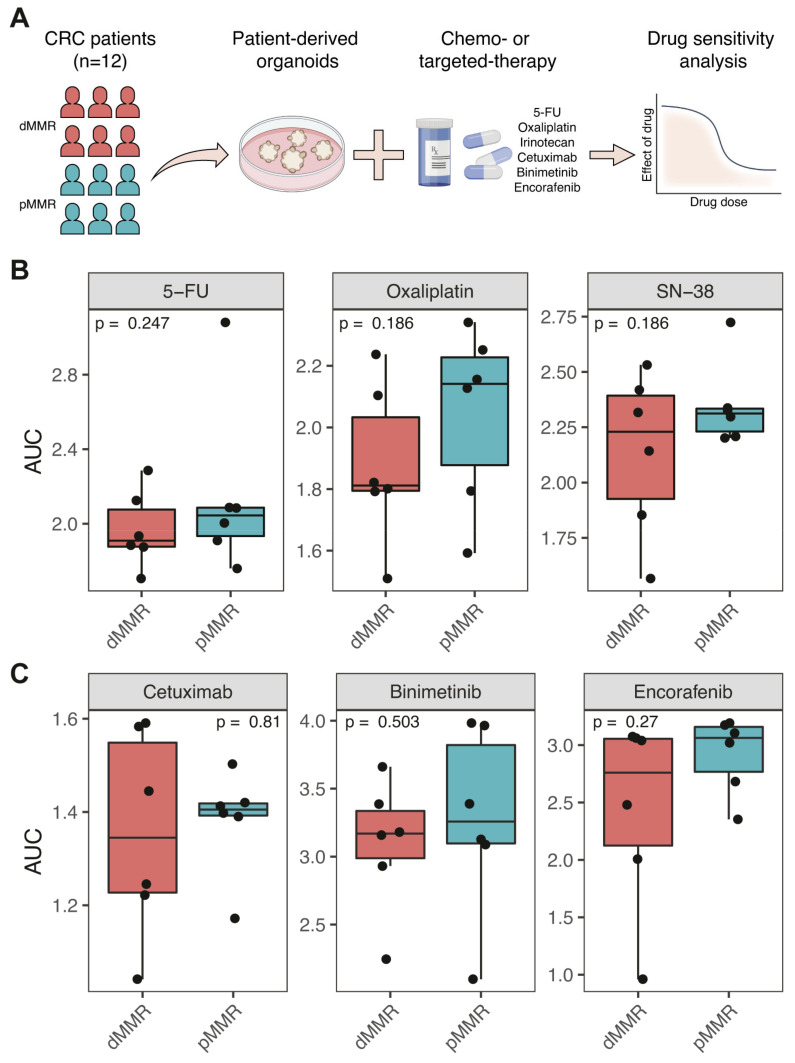
Sensitivity of dMMR versus pMMR PDOs to systemic treatment. (**A**) Schematic overview of the study. Per treatment type, boxplots display the average of each drug response curve (DRC) parameter per organoid line, grouped per mismatch repair (MMR) status, with significance tested using the Mann–Whitney U-test. (**B**) The AUC for chemotherapy (5-FU, oxaliplatin and SN-38 (irinotecan)) are displayed. (**C**) Similarly, the AUCs for targeted treatment (cetuximab, binimetinib and encorafenib) are displayed.

**Figure 4 cancers-13-05434-f004:**
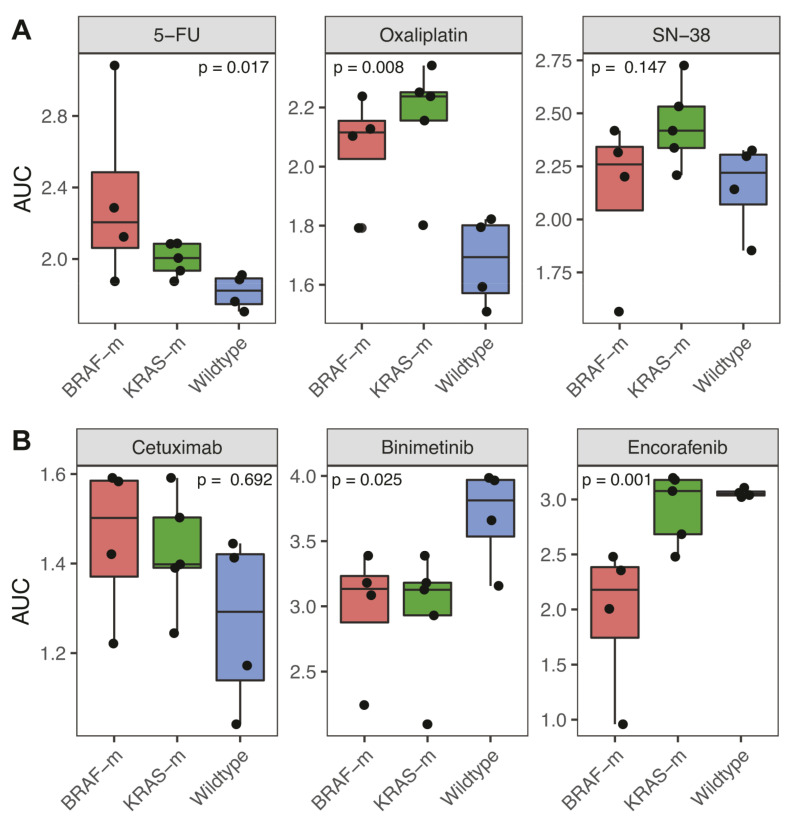
Sensitivity of *BRAF*-mutant versus *KRAS*-mutant versus wildtype PDOs to systemic treatment. Per treatment type (5-FU, oxaliplatin, SN-38 (irinotecan), cetuximab, binimetinib, encorafenib), boxplots display the average of each drug response curve (DRC) parameter per organoid line, grouped per mutational status (*BRAF*-mutant (BRAF-m), *KRAS*-mutant (KRAS-m) or wildtype for *BRAF* and *KRAS* (Wildtype)). (**A**) The AUC for chemotherapy (5-FU, oxaliplatin and SN-38 (irinotecan) are displayed, significance tested using the Kruskal–Wallis test. (**B**) Similarly, the AUCs for targeted treatment (cetuximab, binimetinib and encorafenib) are displayed. Organoid line dMMR4 has both a *KRAS* and *BRAF* mutation, so was included in both categories.

**Table 1 cancers-13-05434-t001:** Characteristics of PDOs.

Organoid	MMR Status (IHC)	*KRAS* Status	*BRAF* Status	Source of Tissue	Primary Tumor Location (Sidedness)
dMMR1	MLH1/PMS2	WT	V600E	Metastatic	Left
				(liver)	
dMMR2	MLH1/PMS2	WT	WT	Metastatic	Left
				(inguinal lymph node)	
dMMR3	MSH2/MSH6	A146T	WT	Metastatic	Right
				(peritoneum)	
dMMR4	MLH1/PMS2	A146T	V600E	Primary	Right
				(synchronous metastatic disease)	
dMMR5	MLH1/PMS2	WT	WT	Primary	Right
				(unknown if synchronous)	
dMMR6	MLH1/PMS2	WT	V600E	Primary	Right
				(unknown if synchronous)	
pMMR1	Normal	WT	V600E	Primary	Right
				(synchronous metastatic disease)	
pMMR2	Normal	G12A	WT	Metastatic	Right
				(liver)	
pMMR3	Normal	WT	WT	Metastatic	Right
				(liver)	
pMMR4	Normal	G12V	WT	Primary	Unknown
				(unknown if synchronous)	
pMMR5	Normal	G12A	WT	Primary	Right
				(synchronous metastatic disease)	
pMMR6	Normal	WT	WT	Primary	Unknown
				(unknown if synchronous)	

Sidedness of the primary tumor was defined as right-sided (coecum-transverse colon), left-sided (splenic flexure-sigmoid), and rectosigmoid/rectal. Synchronous metastatic disease: if developed metastasis within six months of primary CRC diagnosis. Abbreviations: IHC (immunohistochemistry), dMMR (deficient mismatch repair), MMR (mismatch repair), PDOs (patient-derived organoids), pMMR (proficient mismatch repair), WT (wildtype).

## Data Availability

All data used in the analysis is reported in the Appendix A. Analysis was performed in R (Version 4.0.2) using publicly available packages. The raw drug screen data and used code scripts are available from the corresponding author on reasonable request.

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
