# Peer review of "Mismatch Repair Status in Patient-Derived Colorectal Cancer Organoids Does Not Affect Intrinsic Tumor Cell Sensitivity to Systemic Therapy"

_cancers, 2021, doi:10.3390/cancers13215434_

Round 1

Reviewer 1 Report

The manuscript is well structured and the flow of the text is also very easy to follow. The authors show that the MMR status does not really confer resistance to chemotherapy but the mutational status of BRAF or KRAS does. This is well explained with the series or experiments. The Manuscript looks good averall.

However, it will be interesting if (possible) the authors could check the mutational status of CASC1/LAS1 which is shown to coexpress along with KRAS in KRAS-mutated tumors (https://doi.org/10.3390/cancers13040808). CASC1 is an antiapoptotic gene preventing cell death due to mitotic stress.

Also, do define pMMR as it was done for dMMR. Does pMMR statnd for proficient DNA MMR? 

Reviewer 2 Report

This study utilized the sample collected from patients to establish the MMR–proficient and –deficient organoids. Based on the established organoids, they screened a series of clinical drugs to see if there is a difference in sensitivity to these drugs between MMR-proficient and deficient samples. The article has been prepared well with a statistical standard. However, the limitation of the article is also obvious. The sample size is too sample to make a solid conclusion. Also, it is hard to support the authors’ viewpoint if the samples have different MMR proteins deficiency or genetic mutations. Below are some suggestions and questions about this article.

  1. In this study, six dMMR samples lose different types of MMR protein and have different mutations in the gene. Therefore, the reason why authors cannot see the correlation between MMR status and tumor cells’ intrinsic responsiveness to drugs may be that different types of MMR protein play a different role in the resistance of tumor cells.
  2. “In addition, the variation was minimal between the duplicate assays of each PDO, ranging from -1.6% to 0.6% (Figure S3).” Figure s3 is variation between the biological triplicates instead of the experimental duplicates.

Author Response

Response to Reviewer 2 Comments
Reviewer #2: This study utilized the sample collected from patients to establish the MMR–proficient and –deficient organoids. Based on the established organoids, they screened a series of clinical drugs to see if there is a difference in sensitivity to these drugs between MMR-proficient and deficient samples. The article has been prepared well with a statistical standard. However, the limitation of the article is also obvious. The sample size is too sample to make a solid conclusion. Also, it is hard to support the authors’ viewpoint if the samples have different MMR proteins deficiency or genetic mutations. Below are some suggestions and questions about this article.
Point 1: In this study, six dMMR samples lose different types of MMR protein and have different mutations in the gene. Therefore, the reason why authors cannot see the correlation between MMR status and tumor cells’ intrinsic responsiveness to drugs may be that different types of MMR protein play a different role in the resistance of tumor cells.
Response 1: Within the PDO cohort analyzed, we do find statistically significant relationships of BRAF and KRAS mutations with therapy response, but we do not find such a correlation with dMMR when analyzed as a single CRC entity. We agree with the reviewer that it is possible that specific subsets of dMMR-causing (epi-) genetic alterations may show a correlation with therapy response, but our study was not designed to detect such dMMR-subgroup-specific signals. A deficient MMR can arise due the hypermethylation of one of the MMR proteins (sporadic) or due to a germline mutation in MMR genes (inherited; Lynch-syndrome). Indeed, a much larger cohort will be required to investigate the role of different types of MMR-causing (epi-) genetic alterations in CRC therapy resistance. This goes beyond the scope of the current manuscript.
Point 2: “In addition, the variation was minimal between the duplicate assays of each PDO, ranging from -1.6% to 0.6% (Figure S3).” Figure s3 is variation between the biological triplicates instead of the experimental duplicates.
Response 2: We have adjusted the Bland-Altman plot in Supplementary Figure S3; now demonstrating the variation between the duplicate assays. The y-axis of the plot is the difference between technical replicates (log-transformed). The X-axis is the average (mean) between technical replicates (log-transformed). We have adjusted the figure legend and made textual changes in the manuscript at line 197.

Reviewer 3 Report

This is a well performed study which open new horizons in the treatment of patients with metastastic colorectal cancer.

As all good studies the conclusions of the Authors determine more questions than aswers.

Do only Kras and BFRA mutations play a role in resistance to chemotherapy? This resistance is common for all chemotherapy regimens?

Probably the Authors should add this matter in their discussion.

I congratulate the Authors for their work which has required signficant efforts, time and dedication
